# MED15 prion-like domain forms a coiled-coil responsible for its amyloid conversion and propagation

Cristina Batlle[1], Isabel Calvo [2], Valentin Iglesias [1], Cian J. Lynch[2], Marcos Gil-Garcia[1], Manuel Serrano [2,3] & Salvador Ventura[1✉]

A disordered to β-sheet transition was thought to drive the functional switch of Q/N-rich prions, similar to pathogenic amyloids. However, recent evidence indicates a critical role for coiled-coil (CC) regions within yeast prion domains in amyloid formation. We show that many human prion-like domains (PrLDs) contain CC regions that overlap with polyQ tracts. Most of the proteins bearing these domains are transcriptional coactivators, including the Mediator complex subunit 15 (MED15) involved in bridging enhancers and promoters. We demonstrate that the human MED15-PrLD forms homodimers in solution sustained by CC interactions and that it is this CC fold that mediates the transition towards a β-sheet amyloid state, its chemical or genetic disruption abolishing aggregation. As in functional yeast prions, a GFP globular domain adjacent to MED15-PrLD retains its structural integrity in the amyloid state. Expression of MED15-PrLD in human cells promotes the formation of cytoplasmic and perinuclear inclusions, kidnapping endogenous full-length MED15 to these aggregates in a prion-like manner. The prion-like properties of MED15 are conserved, suggesting novel mechanisms for the function and malfunction of this transcription coactivator.

[1] Institut de Biotecnologia i Biomedicina and Departament de Bioquímica i Biologia Molecular, Universitat Autónoma de Barcelona, Bellaterra, Spain. [2] Institute for Research in Biomedicine (IRB Barcelona), The Barcelona Institute of Science and Technology, Barcelona, Spain. [3] Catalan Institution for Research and Advanced Studies (ICREA), Barcelona, Spain. ✉email: salvador.ventura@uab.es

Prions were first described as protein-only infectious agents in the context of mammalian neurological disorders[1]. Nevertheless, increasing evidence indicates that prion-like conformational conversion is not always deleterious; instead, it can be exploited for beneficial purposes[2]. The best characterized nonpathogenic prions are those of yeast, where they provide increased fitness in fluctuating environments[3,4], mainly by regulating transcription and translation[5]. Yeast prions are modular proteins, and their conformational promiscuity is encoded at their prion domains (PrDs), which are both sufficient and necessary for prion conversion[6]. PrDs are long and disordered sequences of low complexity, often enriched in glutamine and/or asparagine (Q/N) residues. Polypeptides containing similar Q/N-rich prion-like domains (PrLDs) have been identified in other organisms, including humans, and they are generically named prion-like proteins[2].

The aggregated state of yeast prions is macroscopically indistinguishable from that of pathogenic prions and amyloid proteins involved in neurodegenerative diseases[7]. Thus, it has been assumed that they all form amyloids through a common mechanism, which involves misfolding of the native structure, either folded or disordered, into β-sheet aggregation-prone conformations[8]. However, this generic and uncontrolled misfolding mechanism was difficult to reconcile with the evidence that, in contrast to pathogenic amyloids, the structural transitions of functional prions should be regulated. Fiumara and co-workers demonstrated that yeast Q/N-rich PrDs have an intrinsic propensity to form coiled-coils (CC), and they proposed that CC-based structural changes might explain better the physiological conformational switches of functional Q/N-rich prions than stochastic misfolding[9].

As yeast PrDs, polyQ tracts can adopt CC structures[10]; however, in this case, CC-mediated conformational changes might be pathogenic. Several rare hereditary neurodegenerative diseases are originated when Q stretches exceed a critical length, and their severity increases with polyQ expansion[11,12]. These extensions would result in longer CC-prone helices, with a stronger polymerization propensity, which might ultimately cause their aggregation and the formation of the recurrent protein deposits observed in these diseases[9].

Although CC-mediated aggregation provides a plausible mechanism for functional transitions in yeast PrDs and for the pathogenesis of polyQ-expansion disorders, it remains to be demonstrated that the aggregation of human prion-like proteins also accommodates to the CC model. Therefore, we undertook a bioinformatic analysis of the human proteome, which revealed that a significant proportion of human PrLDs indeed contained sequences of high CC propensity, which, in most cases, overlapped with Q-rich regions. This subgroup of prion-like proteins works in the regulation of transcription, and it comprises key transcription coactivators, including Mediator complex (Mediator) subunits 12 and 15 (MED12 and MED15).

Mediator is a large multi-protein complex that regulates most, if not all, enhancer-driven gene transcription[13–15]. Mediator is highly conserved among eukaryotes and consists of four modules: head, middle, tail, and the CDK8 kinase module[16]. The head and middle modules execute recruitment to the promoter regions, while the tail module mediates protein interactions with the transcription regulators at enhancers, and the CDK8 module is involved in RNA Polymerase II (Pol II) recruitment and release[15]. The tail module consists of 7 subunits, one of them being MED15, which, as described above, we identified as a CC forming prion-like protein.

MED15 is located in the cell nucleus, and its knockdown causes slow growth and reduced transcriptional activation[17]. MED15 is overexpressed in a wide range of human cancers: castration-resistant prostate cancer, head, and neck squamous cell carcinoma,

hepatocellular carcinoma, breast cancer, renal cell carcinoma, and testicular germ cell tumors[18–25]. Patients with MED15 over-expression in tumor tissues exhibit a more aggressive phenotype, associated with significantly shorter survival time[18]. Consequently, MED15 may serve as a prognostic marker as well as a potential therapeutic target in cancer.

The MED15 subunit was first discovered in yeast as Gal11[26] and later on renamed as yMED15[27]. yMED15 is involved in a variety of biological processes, such as the expression of galactose-inducible genes, and it is an interaction hub of many transcription factors (TFs)[28]. Interestingly, yMED15 contains a central Q-rich region reported to form amyloid-like intracellular inclusions spontaneously in yeast, whereas the aggregation of the full-length yMED15 occurs under stress conditions[29]. The aggregation of yMED15 provokes the loss of the entire tail module, reducing transcription levels, which has been suggested to be an epigenetic mechanism for transcription regulation, similar to those of classical yeast prions[29].

Human MED15 (MED15) has a Q-rich region, consisting of discontinuous polyQ tracts at its N-terminus, which maps to its PrLD and is responsible for the predicted CC propensity of this domain (Fig. 1A). yMED15 and MED15 have low sequence identity and different distribution of their Q-rich regions; still, this kind of low complexity sequences have been reported to exert similar functions in orthologs without a strict positional requirement[10]. Therefore, the MED15 prion-like domain (MED15-PrLD) might possess cryptic amyloid and prion-like features yet to be discovered.

Here, we provide experimental evidence that the MED15-PrLD forms a dimer with CC structure. This domain bears the ability to self-assemble into Congo Red (CR) positive, β-sheet enriched amyloid fibrils, displaying concentration-dependent aggregation kinetics, and self-seeding activity. Moreover, as in functional yeast prions, a GFP globular domain adjacent to the PrLD maintains its fold and activity in the amyloid state. We demonstrate that it is the CC structure that regulates MED15-PrLD amyloid formation, its chemical or genetic disruption abolishing aggregation. Expression of MED15-PrLD in human cells promotes the formation of cytoplasmic and perinuclear inclusions, kidnaping the endogenous full-length protein to these aggregates in a prion-like manner.

Overall, this study reveals that amyloid formation by human prion-like proteins can follow the CC-mediated model reported in yeast and that the amyloid and prion-like properties of MED15 are conserved across species, which suggests novel mechanisms for MED15 function and malfunction.

## Results

### Human PrLDs with predicted coiled-coil domains overlap with polyQ tracts.
Coiled-coils (CC) are formed between proteins that contain repeats of seven amino acids (a-b-c-d-e-f-g) in which hydrophobic residues often occupy positions a/d to form a hydrophobic layer between the coiling helices; these heptad repeats are often discontinuous[30,31]. CC regions are over-represented in yeast prions, overlapping with their Q/N-rich prion domains, suggesting that the CC structure might represent a defining feature of these domains in yeast[9]. This observation prompted us to assess whether human prion-like proteins might share the same architecture.

First, we performed a prion-like amino acid composition (PLAAC) analysis to identify human prion-like proteins and their respective PrLDs at the proteome level. PLAAC is an algorithm that allows the identification of polypeptide regions with a composition similar to that of a set of well-characterized yeast PrDs[32]. We obtained a list of 193 reviewed human genes encoding for proteins with PrLDs (Supplementary Data 1).

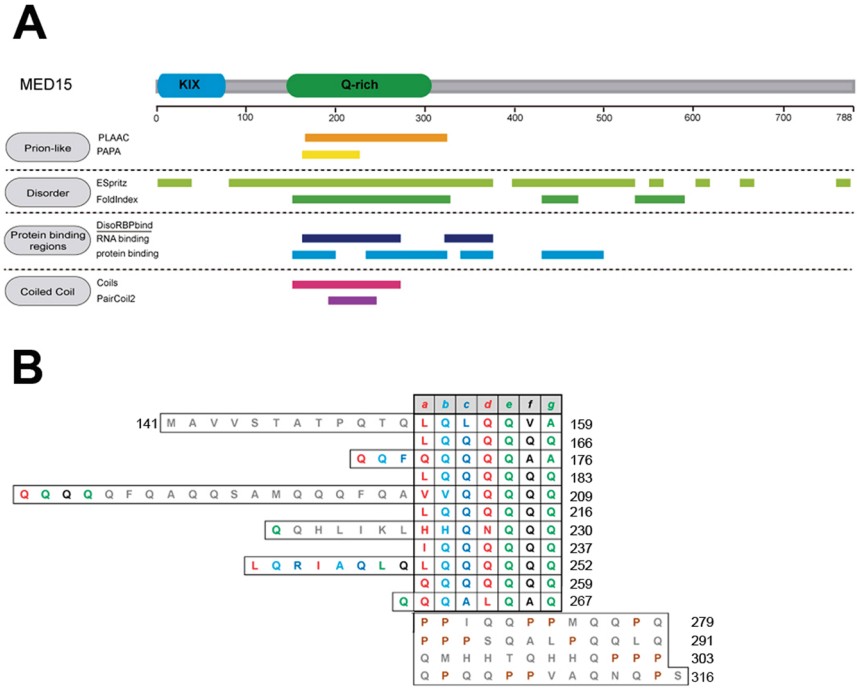

**Fig. 1 MED15 contains an N-terminal Q-rich coiled-coil PrLD. A** MED15 diagram showing the location of the KIX (blue) and Q-rich (green) regions. Prediction of prion-like domains using PLAAC[32] and PAPA[72] predictors, disorder regions using ESpritz[38] and FoldIndex[37] predictors, protein binding regions using DisoRBPbind[39] predictor, and coiled-coil regions using COILS[33] and PairCoil2[70] predictors are shown below the diagram. **B** Schematic diagram of the heptad repeats (a-b-c-d-e-f-g) predicted by COILS[33] (14-residue window) of MED15-PrLD (residues 141–316). Gray residues indicate COILS score <0.2.

We analyzed which of the identified PrLDs might fold into a CC using COILS[33], an algorithm that detects CC heptad repeats in primary sequences (Supplementary Data 1). We found that 12% of human PrLDs ($n = 22$) were predicted to have CC regions. In all, 86% of these proteins ($n = 19$) is involved in transcriptional regulation (Supplementary Data 1). Accordingly, a Gene Ontology (GO) analysis of the 22 candidates[34] indicated a significant enrichment in the following ontologies: (i) transcription initiation from RNA polymerase II promoter in biological process, (ii) transcription coactivator activity in molecular function, and (iii) nucleus and nucleoplasm in cellular component (Supplementary Data 1). Importantly, 90% of the predicted CC regions ($n = 20$) correspond to or overlap with polyQ tracts in the PrLDs (Supplementary Data 1 and Fig. S1), which is consistent with the observation that polyQ proteins are functionally biased towards transcriptional regulation[10,35]. These 20 prion-like candidates include proteins whose polyQ expansion is connected to autosomal dominant cerebellar ataxia, such as the TATA-box-binding protein (TBP) and ATXN1[11], as well as the CREB-binding protein (CBP), a transcriptional coactivator necessary for the survival of many neurons which is recruited into aggregates in polyQ diseases[36]. Overall, our analysis indicates that, as it occurs in yeast prions, a significant fraction of human prion-like proteins might contain CC that overlap with polyQ tracts at their PrLDs. Proteins displaying these features include important transcriptional coactivators like CBP, TBP, KMT2D, FOXP2, NCOA2, NCOA3, MAML2, MED12, and MED15.

Apart from those involved in disease, the amyloid and prion-like properties of the PrLDs identified in these transcriptional regulators remain to be experimentally demonstrated. However, the yMED15 polyQ can form amyloid-like aggregates on its own[29]. Although human and yeast MED15 proteins display very low sequence identity (12%), we decided that it was worthwhile to explore whether the identified human MED15-PrLD can access

an amyloid state and propagate this conformation in a prion-like manner.

**MED15 contains an N-terminal Q-rich coiled-coil PrLD.** MED15 is highly enriched in Gln residues (Q, 20% of residues), especially at its N-terminus, with multiple short polyQ tracts (of 2–16 Qs; Fig. 1). It is precisely this N-terminal Q-rich region that is identified as a PrLD by PLAAC (Fig. 1A and Supplementary Data 1).

PrLDs are thought to be intrinsically disordered in their monomeric states, in such a way that their self-assembly can occur spontaneously, without an energy-dependent conformational change[2]. The MED15-PrLD is consistently predicted to be disordered by FoldIndex and ESpritz predictors (Fig. 1A)[37,38]. Disordered regions can be cryptic protein binding sites that, upon interaction, undergo a disorder to order transition; DisoRDPbind is an algorithm aimed to predict such transitions, scoring putative associations between the disordered region of interest and RNA, DNA, or protein[39]. MED15-PrLD was predicted to interact with both RNA and protein (Fig. 1A), suggesting that it is susceptible to experiment conformational switches. Indeed, polyQ tracts are known to mediate protein–protein interactions (PPI) and multimerization, which can lead to the formation of homomeric or heteromeric CC structures[10]. Accordingly, the MED15 N-terminal Q-rich region is predicted to adopt a CC conformation (Fig. 1A and Supplementary Data 1) and has been shown to exhibit α-helical structure when fused to a globular domain[40]. This CC domain is discontinuous and consists of three consecutive CC regions (Fig. S2). As expected, in the MED15-PrLD CC model, almost all "a" and "d" positions in the helix are occupied by hydrophobic amino acids or Gln, which acts as an ambivalent hydrophobe (Fig. 1B)[41]. The N-terminal Q-rich segment is the less conserved region in MED15 vertebrate orthologs, with significant differences in the number, length, and

location of polyQ tracts among species; however, in all cases, it is predicted to form CC (Fig. S2), suggesting that this region is structurally conserved, likely, because of its functional significance.

Overall, the bioinformatics analysis of MED15 suggests that we are in front of a prion-like protein with a disordered N-terminal Q-rich PrLD in its monomeric state, but with the ability to undergo a transition to a dimeric or higher-order CC structure.

**MED15 interactors are predicted to have coiled-coil domains.** Recently, a list of Mediator complex interactors was identified by performing FLAG-MED15 immunoprecipitation in mice[42]. Therefore, at least a fraction of the identified binders may contact MED15 directly. All mouse MED15 potential interactors display >75% sequence identity with their human homologs (Supplementary Data 2). In the light of the predicted propensity of MED15 to form CC, we examined the CC propensity of their putative interactors using COILS[33]. Interestingly enough, 50% of these candidates resulted in positive predictions, indicating that CC-containing proteins are overrepresented among MED15 binders (Supplementary Data 2). Not surprisingly, a GO term enrichment analysis indicated that this subset of proteins is preferentially located at the nucleoplasm, acting as transcription coactivators and with major involvement in Pol II transcription (Supplementary Data 2). Importantly, among these polypeptides, we found four of the CC-containing prion-like proteins identified in the previous section: NCOA2, MAML2, CBP, and MED12, which suggests that their PrLDs compositional bias and propensity to populate CC conformations might result in their eventual interaction with MED15. Indeed, an analysis with the STRING protein interactions database[43], indicates that all these five proteins are closely associated (Fig. S3; PPI enrichment $p$-value: $7.5 \times 10^{-08}$).

**Soluble MED15-PrLD characterization.** Long polyQ stretches are commonly followed by polyproline (polyP) tracts at their C-terminus. These repeats stabilize and stop the CC region[10,44], which suggests that the two tracts form a functional/structural unit and that the expression of the polyQ alone may not recapitulate the physiological context of the sequence. In MED15, a Pro-rich segment is also adjacent to the Q-rich region (Fig. 1B), and both stretches are predicted to be part of the PrLD. The presence of Pro residues at the C-terminus of polyQ tracts is also frequent in MED15 orthologs (Fig. S4).

Most proteins with polyQ tracts are modular, and these stretches are in close vicinity to globular domains. Indeed, these low complexity regions are difficult to express recombinantly and to purify in the absence of a globular partner that modulates their inherent aggregation propensity[45]. The MED15-PrLD is flanked at its N-terminus by a globular KIX domain (Fig. 1A), a feature conserved in MED15 orthologs (Fig. S4).

With the above considerations in mind, we expressed the MED15-PrLD domain, containing both Q-rich and Pro-rich regions, fused at the C-terminus of GFP, which would act as a functionally traceable globular domain, hereinafter named as GFP-MED15CC. This construct should allow us to characterize the soluble, and eventually, the aggregated states of MED15-PrLD in a mimic of its sequential framework.

As intended, GFP-MED15CC was purified from the soluble cellular fraction. The GFP globular domain integrity was evaluated by monitoring the GFP absorption and fluorescence emission spectra (Fig. 2A, B). GFP-MED15CC spectral properties were identical to those of GFP alone, with an absorption maximum at 490 nm and an emission fluorescence maximum at 510 nm. In order to verify that GFP-MED15CC was not aggregated in the soluble state, we used synchronous light scattering (Fig. 2C). Neither GFP nor GFP-MED15CC showed a significant increase in light scattering signal, compared to the buffer alone.

The GFP-MED15CC secondary structure was assessed by far-UV circular dichroism (CD; Fig. 2D). GFP alone showed a β-sheet spectrum with a characteristic minimum at 218 nm. In contrast, the GFP-MED15CC spectrum exhibited two minima at 208 and 222 nm, characteristic of α-helices. The deconvolution of the spectra using the K2D program[46] indicated a predominant α-helix conformation for GFP-MED15CC (100%) and β-sheet for GFP (78%). Thus, the strong α-helix signal in GFP-MED15CC completely masks the signal of GFP, something that is characteristic of globular domains-CC fusions[47].

As described above, MED15-PrLD is predicted to be intrinsically disordered in its monomeric state. The fact that we observed a predominant α-helical secondary structure by far-UV CD is consistent with this region adopting a CC. This necessarily implies that, in solution, GFP-MED15CC acquires quaternary structure. GFP-MED15CC is 458 residues long, with an expected MW of 52 kDa. Size exclusion chromatography (SEC) and native PAGE were performed to evaluate the oligomerization state of GFP-MED15CC (Fig. 2E, F). Its SEC elution pattern corresponded to a protein size of ~100 kDa (elution at 12.65 ml), indicating that the protein was a dimer (Fig. 2E). In contrast, GFP (246 residues, MW 28 kDa) was eluted as a monomer (elution at 15 ml). In agreement with the SEC data, GFP-MED15CC migrates as a unique band of ~100 kDa in a native PAGE, located between the monomeric and dimeric forms of bovine serum albumin (BSA) used as control (Fig. 2F). Besides, dynamic light scattering (DLS) analysis revealed major populations with average diameters of 13 and 7 nm for GFP-MED15CC and GFP, respectively, further supporting that the MED15 CC domain drives the formation of a dimeric structure.

In conclusion, in solution, GFP-MED15CC is a dimer sustained by a CC interaction, where the adjacent globular domain retains its integrity and function.

**MED15-PrLD aggregates into β-sheet enriched amyloid-like fibrillar structures.** Both PrLDs and polyQ tracts are well known for their ability to aggregate into amyloid β-sheet conformations[45,48]. Therefore, we assessed if GFP-MED15CC aggregates spontaneously using synchronous light scattering, GFP fluorescence, far-UV CD, and infrared spectroscopy (ATR-FTIR; Fig. 3A–D). GFP was used as a control in all these assays.

The proteins were incubated at 5 μM (0.25 mg/ml) for 24 h at 37 °C under quiescent conditions, and the light scattering of the correspondent solutions measured (Fig. 3A, B). In contrast to GFP, GFP-MED15CC exhibited a substantial increase in the light scattering signal upon incubation (Fig. 3A). GFP fluorescence was examined in the samples' supernatant after their centrifugation (see Methods section). In agreement with the light scattering data, the fluorescence of the GFP-MED15CC in the soluble fraction decreased significantly upon incubation, whereas that of GFP remained mostly the same (Fig. 3B). Together, these results demonstrate that GFP-MED15CC aggregates within 24 h.

We monitored the secondary structure of incubated GFP-MED15CC solutions by far-UV CD and fourier transform infrared (FTIR; Fig. 3C, D). GFP-MED15CC suffered a conformational change from the initial α-helical to a β-sheet structure after 24 h of incubation, displaying a spectrum minimum at 218 nm, similar to that of GFP (Fig. 3C). Prolonged incubation for 4 days resulted in a weak β-sheet signal, indicating that a significant proportion of the protein was aggregated out of the solution, becoming undetectable by CD. The conformation of

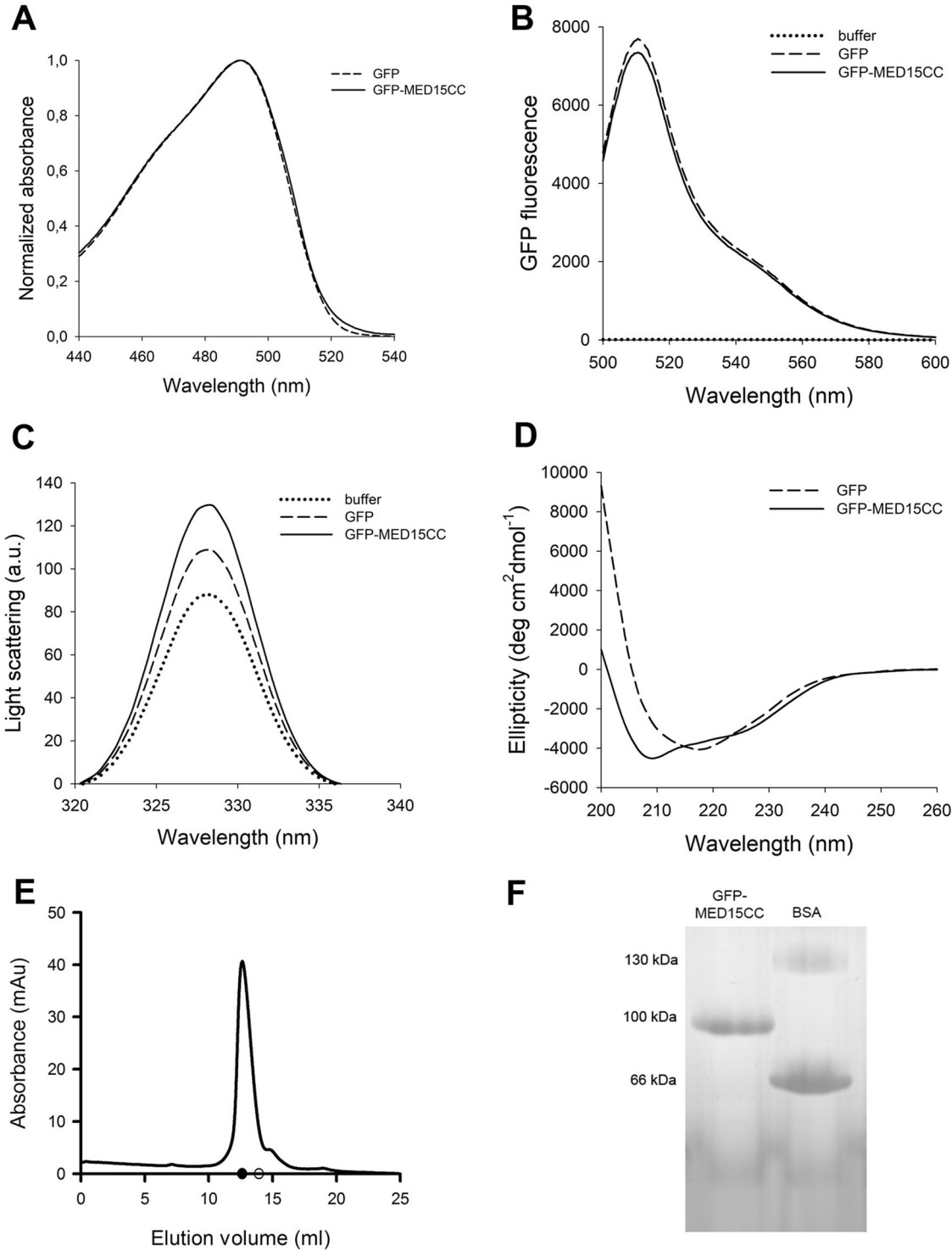

**Fig. 2 MED15-PrLD soluble characterization. A** Absorbance, **B** fluorescence, **C** light scattering, and **D** circular dichroism of 5 μM GFP and GFP-MED15CC. **E** GFP-MED15CC size exclusion chromatography elution profile. Filled circle and empty circle in the x axis indicates 100 and 50 kDa position, respectively. **F** 12.5% blue native PAGE of GFP-MED15CC and bovine serum albumin (BSA) at 0.5 mg/ml. All assays are performed in 20 mM Tris pH 7.5 and 150 mM NaCl.

control GFP remained unaffected after 4 days of incubation (Fig. S5). To further confirm the putative β-sheet secondary structure of incubated GFP-MED15CC, we analyzed its FTIR spectrum in the amide I region (1700–1600 cm$^{-1}$), which corresponds to the absorption of the main chain carbonyl group and is sensitive to the protein conformation. Deconvolution of the FTIR spectrum of incubated GFP-MED15CC resulted in a major peak at 1625 cm$^{-1}$, which is typically attributed to the presence of intermolecular β-sheet structure (Fig. 3D), in good agreement with the CD data.

Next, we evaluated whether GFP-MED15CC aggregates were arranged into an amyloid-like architecture. GFP-MED15CC emits green fluorescence; hence, we could not use Thioflavin-T[49]. Instead, we used a CR precipitation assay since this dye

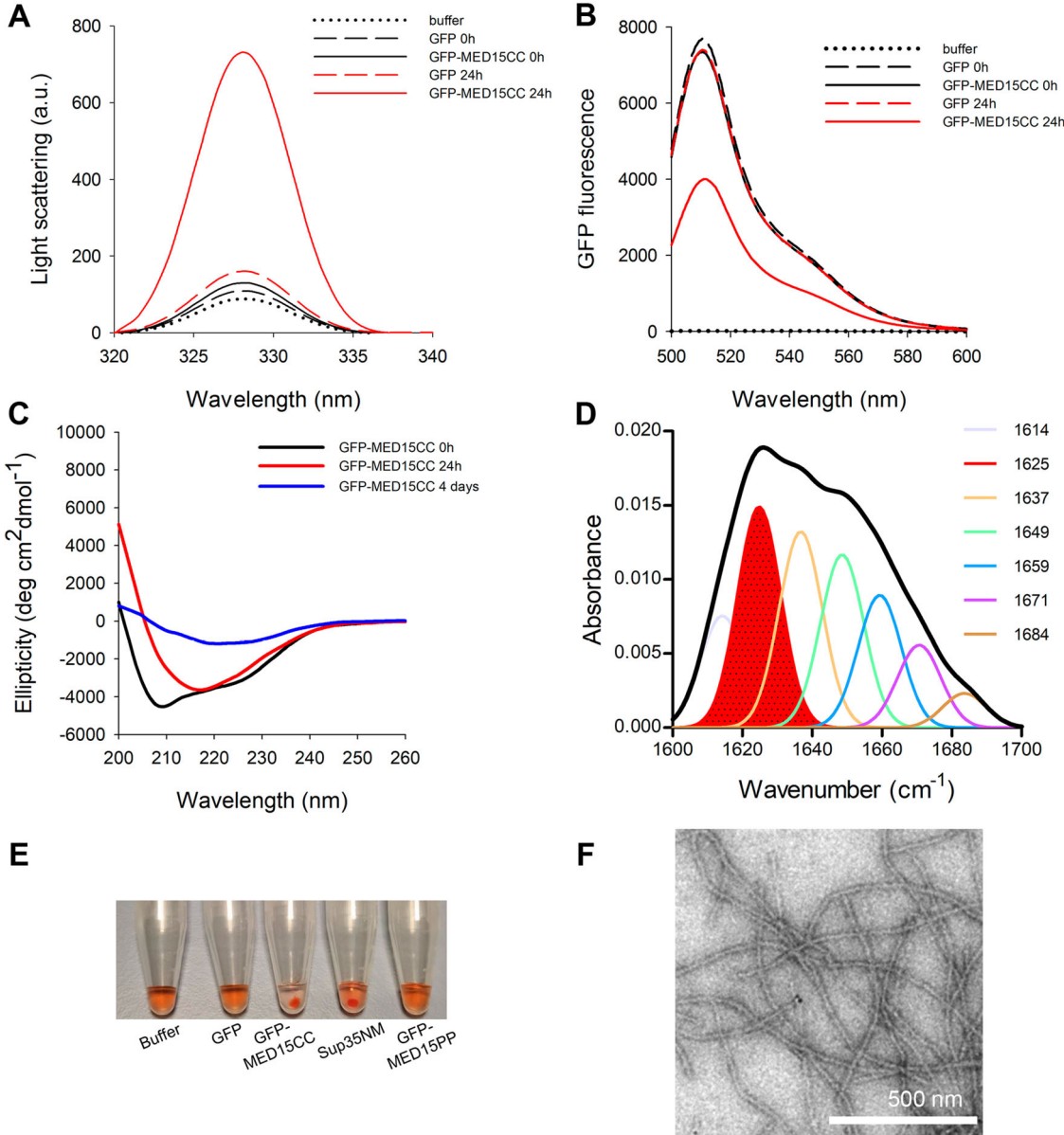

**Fig. 3 MED15-PrLD aggregates into β-sheet enriched amyloid-like fibrillar structures. A** Light scattering, **B** GFP fluorescence (supernatant fraction) and **C** circular dichroism of soluble ($t = 0$ h, black) and aggregated ($t = 24$ h, red, or $t = 4$ days, blue) 5 µM GFP-MED15CC or GFP. **D** FTIR of aggregated ($t = 24$ h) 5 µM GFP-MED15CC. **E** Congo Red precipitation of incubated ($t = 9$ h) 5 µM GFP, GFP-MED15CC, GFP-MED15PP, and 10 µM Sup35NM. **F** Transmission electron microscopy image of aggregated ($t = 9$ h) 5 µM GFP-MED15CC. All assays are performed in 20 mM Tris pH 7.5 and 150 mM NaCl.

deposits on top of amyloid fibrils[50]. We incubated CR with buffer only, or with GFP, as negative controls, and with Sup35NM yeast prion amyloid fibrils as a positive control[5]. After 1 h of incubation, only GFP-MED15CC and Sup35NM formed a red pellet, suggesting that MED15CC aggregates are amyloid-like (Fig. 3E). Next, we addressed the morphological features of GFP-MED15CC aggregates by transmission electron microscopy (TEM) to further confirm its amyloidogenic nature (Fig. 3F). Negative staining showed long, thin, and unbranched amyloid-like fibrillar structures, without any significant accumulation of amorphous material. The fibrils exhibited a diameter of 19 ± 4 nm and a length of 2 ± 0.5 µm.

In contrast to pathogenic proteins, a remarkable property of yeast PrDs is that their assembly into amyloids does not necessarily imply the misfolding of the attached globular domains, which might keep their native conformation in the aggregated state[51]. In the same manner, GFP-MED15CC fibrils were green fluorescent, indicating the maintenance of the native GFP fold within the fibril phase (Fig. S6).

Overall, these results indicate that the PrLD of the initially dimeric GFP-MED15CC protein promotes the spontaneous assembly into supramolecular β-sheet enriched fibrillar amyloid structures in which the adjacent globular domain keeps its functional conformation, all these features being consistent with a prion-like nature.

**MED15-PrLD aggregation kinetics.** In the previous section, we showed that GFP-MED15CC aggregates into amyloid fibrils. To further characterize this reaction, we followed its kinetics, at 5 µM protein concentration and 37 °C without agitation. We monitored the reaction by synchronous light scattering, far-UV CD, and GFP fluorescence (Fig. 4A–D). Upon incubation, GFP-MED15CC showed a progressive increase in light scattering, indicative of protein aggregation (Fig. 4A). Far-UV CD analysis indicated that

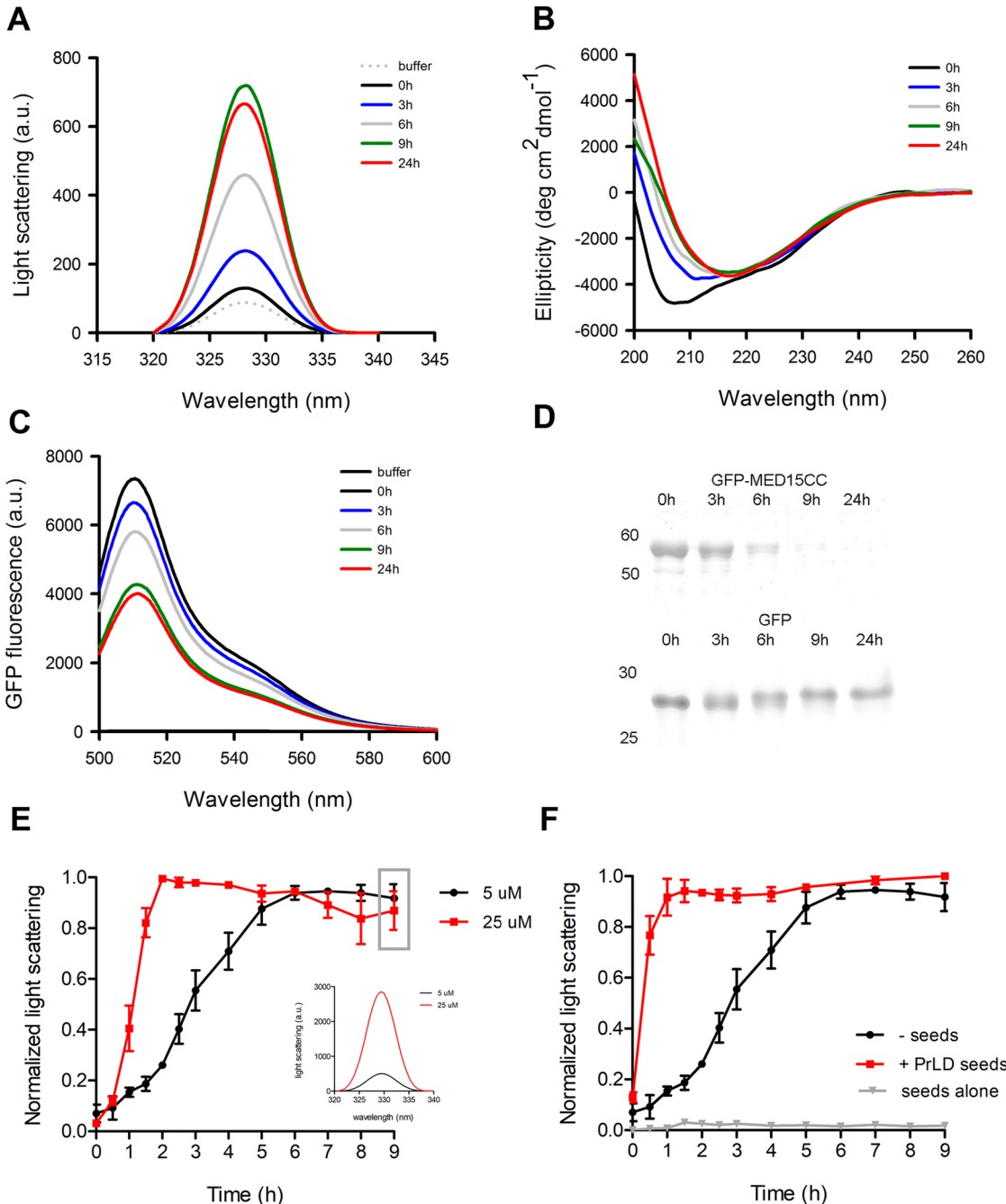

**Fig. 4 MED15-PrLD aggregation kinetics characterization. A** Light scattering, **B** circular dichroism, and **C** GFP fluorescence (supernatant fraction) of 5 µM GFP-MED15CC incubated at 37 °C at the indicated time points. **D** 12% SDS-PAGE of 5 µM GFP and GFP-MED15CC supernatant fraction after centrifugation at the indicated time points of incubation. **E** Light scattering kinetics of 5 and 25 µM GFP-MED15CC at 37 °C. The inlet shows the real scattering intensities at the final time point (9 h, gray rectangle). **F** Light scattering kinetics of 5 µM GFP-MED15CC incubated at 37 °C in the absence or presence of 2% GFP-MED15CC seeds. Kinetics of seeds alone is shown as control. All assays are performed in 20 mM Tris pH 7.5 and 150 mM NaCl.

this is accompanied by a conformational change from an α-helical to a β-sheet secondary structure (Fig. 4B). GFP-MED15CC sample centrifugation at the same time points confirms a gradual decrease in the amount of soluble protein, with a concomitant decrease of GFP fluorescence in the supernatant (Fig. 4C, D). All the techniques converge to indicate that 9 hours of incubation are enough to reach the plateau phase of the aggregation reaction. In contrast, the parallel GFP control did not show signs of aggregation by any of the used techniques (Figs. 3A–C and 4D).

Aggregation into amyloids responds to a second or higher reaction and thus is, typically, very sensitive to the protein

concentration. We evaluated if this is the case for GFP-MED15CC. We monitored GFP-MED15CC aggregation kinetics at 5 and 25 µM by synchronous light scattering and observed a clear acceleration and signal increase at the higher concentration, as expected for a canonical amyloid protein (Fig. 4E).

Many amyloids, and specially prion-like proteins, have the ability to self-propagate by seeding the aggregation of their soluble counterparts[52]. In this way, the presence of small amounts of preformed amyloid fibrils usually accelerates the protein aggregation kinetics, mostly by reducing or even abrogating the lag phase of the reaction, which corresponds to

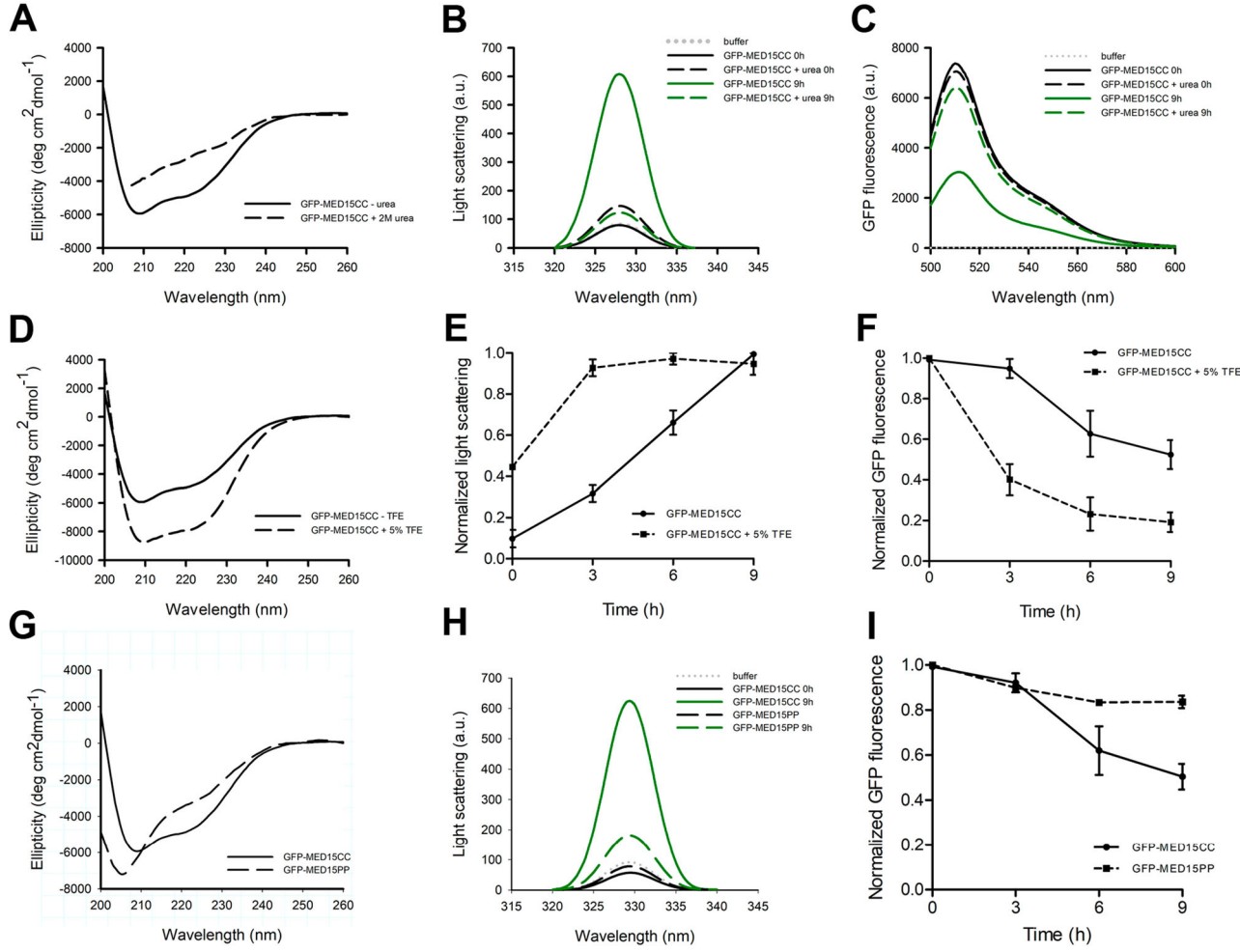

**Fig. 5 MED15-PrLD aggregation is governed by its coiled-coil conformation. A** Circular dichroism, **B** light scattering, and **C** GFP fluorescence (supernatant fraction) of 5 μM GFP-MED15CC in the absence or presence of 2 M urea. **D** Circular dichroism, **E** light scattering kinetics, and **F** GFP fluorescence kinetics (supernatant fraction) of 5 μM GFP-MED15CC in the absence or presence of 5% TFE. **G** Circular dichroism, **H** light scattering, and **I** GFP fluorescence kinetics (supernatant fraction) of 5 μM GFP-MED15CC or GFP-MED15PP. In all cases, aggregation was performed for 9 hours at 37 °C without agitation in 20 mM Tris pH 7.5 and 150 mM NaCl.

the formation of the initial aggregation nuclei. In Fig. 4F we compare the aggregation kinetics of GFP-MED15CC at 5 μM in the absence or presence of 2% (w/w) of its preformed fibrils. It can be observed how the presence of fibrils dramatically accelerates GFP-MED15CC aggregation. This result indicates that GFP-MED15CC amyloid formation follows a nucleation-polymerization mechanism and that, at least in vitro, the MED15-PrLD can propagate its amyloid conformation.

**MED15-PrLD aggregation is governed by its coiled-coil conformation.** Pathogenic polyQ tracts are generally thought to misfold spontaneously into aggregation-prone β-sheet conformations. However, recent evidence suggests that their aggregation depends instead on the population of metastable CC structures[9]. To test whether this mechanism drives GFP-MED15CC amyloid formation, we evaluated its aggregation in the presence of two reagents aimed at weakening or strengthening the CC domain selectively.

In the presence of 2 M urea, the GFP-MED15CC α-helical conformation was significantly disrupted, as assessed by far-UV CD, and the spectrum shifted towards the disordered region (Fig. 5A). Importantly, the control GFP conformation was not affected by this denaturant (Fig. S7A). We analyzed the GFP-MED15CC aggregation propensity in the presence or absence of

2 M urea by synchronous light scattering and GFP fluorescence (Fig. 5B, C). GFP-MED15CC did not show any signs of aggregation in the denaturing agent's presence, indicating that CC disruption prevented protein aggregation. It is important to point out that, as expected, the integrity of the GFP moiety of GFP-MED15CC was not affected by urea because the protein remained fluorescent. The control GFP did not show any aggregation with, or without, urea (Fig. S7B, C).

TFE is a reagent used to enhance or stabilize α-helical conformations. However, the presence of >20% TFE might induce non-native α-helices, blurring any physiologically relevant interpretation of the data; to avoid this effect, we used an unusually low concentration of this cosolvent in our experiments. In the presence of 5% TFE, the CC of GFP-MED15CC was significantly enhanced as observed by far-UV CD (Fig. 5D), whereas the structure of the control GFP was unaffected (Fig. S7D). Evaluation of GFP-MED15CC aggregation kinetics by synchronous light scattering and GFP fluorescence showed a significant acceleration of the process in the presence of TFE (Fig. 5E, F). Control GFP did not show signs of aggregation with, or without, TFE (Fig. S7E, F).

MED15-PrLD contains discontinuous CC segments, accounting for a total of 11 heptad repeats, according to the COILS predictor (Fig. 1B). To confirm the predominant role of MED15 CC on the PrLD aggregation, we used structure-guided

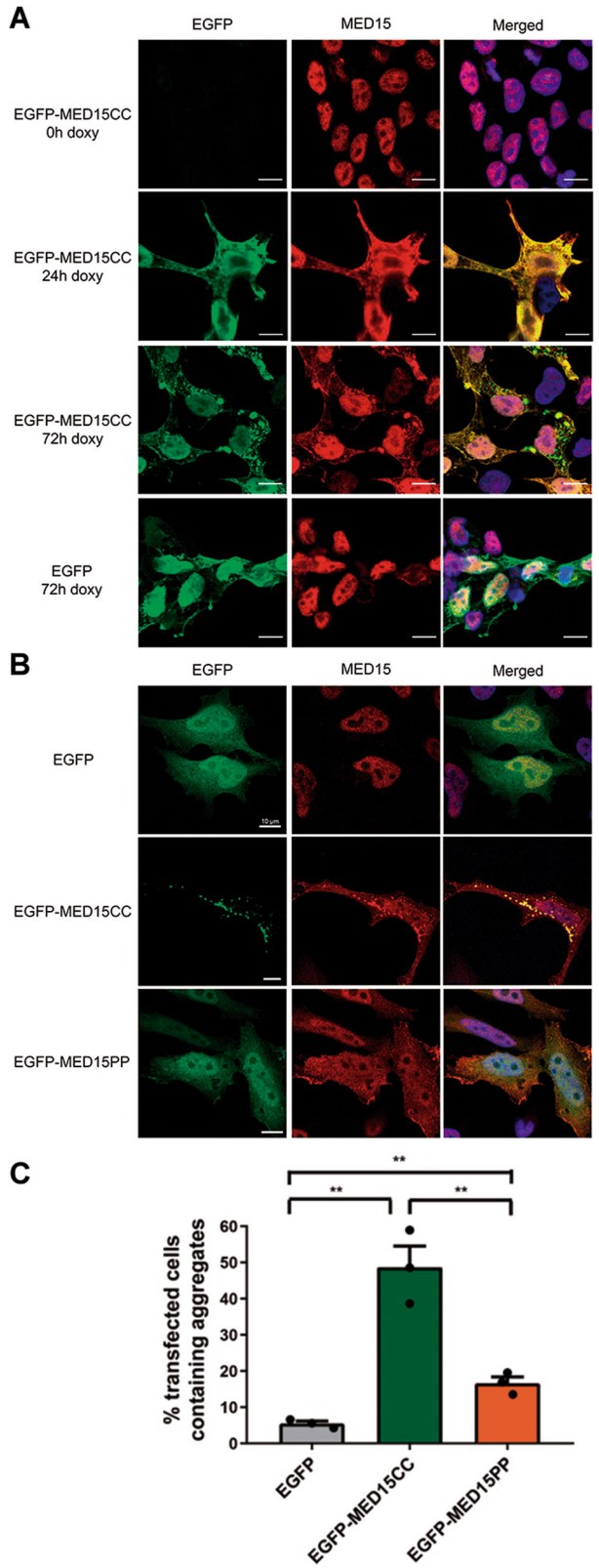

**Fig. 6 MED15-PrLD forms insoluble cytoplasmic inclusions in mammalian cells. A** Cellular localization by immunofluorescence of EGFP or EGFP-MED15CC in HEK293T cells after addition of doxycycline (doxy) for 0, 24, or 72 h. **B** Cellular localization by immunofluorescence of EGFP, EGFP-MED15CC, and EGFP-MED15PP in HeLa cells after expression for 24 h. In both **A** and **B**, cells were stained with MED15 antibody (red) to evaluate the effect of transfection in the endogenous mediator subunit, and with DAPI (blue) as nuclear marker. **C** Graph of percentage transfected cells containing cytoplasmic of perinuclear aggregates after EGFP, EGFP-MED15CC, and EGFP-MED15PP expression in HeLa cells. Dots represent the mean of independent triplicates.

We first analyzed GFP-MED15PP secondary structure in solution by far-UV CD. The variant exhibited a minimum at 205 nm correspondent to a random coil conformation, although a small shoulder at 222 nm was still evident. In any case, the spectrum confirmed that the CC fold was significantly destabilized/disrupted by the introduced amino acid changes (Fig. 5G). Consistent with this observation, the GFP-MED15PP aggregation propensity was clearly reduced, relative to the GFP-MED15CC protein, as evidenced by the synchronous light scattering and GFP fluorescence signals after 9 h of incubation at 37 °C (Fig. 5H, I). Indeed, when incubated under the same conditions that GFP-MED15CC, the mutant protein does not bind to CR (Fig. 3E).

All in all, these results indicate that its CC conformation mediates MED15-PrLD amyloid formation and that disruption of this structure significantly reduces its aggregation potential. This aggregation mechanism is consistent with the one described for the functional switch of Q/N-rich yeast prions[9].

**MED15-PrLD forms insoluble cytoplasmic inclusions in mammalian cells and kidnaps the endogenous protein.** In the previous sections, we have shown that MED15-PrLD forms amyloids and that the CC is a central player in this reaction. To determine whether MED15-PrLD may aggregate in a cellular context, we stably transfected HEK293T cells with lentiviral vectors containing EGFP-tagged MED15-PrLD (EGFP-MED15CC) or EGFP, as a control, and expressed the two proteins using a doxycycline-inducible system (TetO).

MED15 is a component of the Mediator tail module, and, as expected, it displays a diffuse nuclear localization in non-induced cells (Fig. 6A). In all, 24 h after induction of protein expression EGFP-MED15CC formed small aggregates distributed homogenously in the cytoplasm, whereas at 72 hours multiple larger cytoplasmic and perinuclear inclusions where observed, whereas EGFP remained diffusely distributed in the cell (Fig. 6A).

As in the case of HEK293T cells, transfection of HeLa cells with lentiviral vectors containing EGFP-MED15CC or EGFP and expression of the two proteins using the TetO-system resulted in the aggregation of EGFP-MED15CC in the cytoplasm, with EGFP remaining soluble (Fig. S9A).

To demonstrate that the EGFP-MED15CC ability to form intracellular inclusions is expression system independent and that this feature depends on the CC region of the PrLD, we transiently transfected EGFP, EGFP-MED15CC, and the EGFP-MED15PP variant, containing the five disrupting Pro substitutions, in HeLa cells and expressed the three proteins for 24 hours. EGFP-MED15CC expression resulted again in the formation of cytoplasmic and perinuclear inclusions, which occur in ~50% of the transfected cells (Figs. 6B, C and S9B). In contrast, EGFP-MED15PP formed inclusions in only ~15% of the transfected cells (p-value = 0.0063), and they were small (Fig. 6B, C). Inclusion formation is a hallmark of protein aggregation, and aggregated amyloids exhibit protein detergent insolubility. In agreement with their relative in vitro amyloid-forming and

mutagenesis to decrease the CC propensity by introducing Pro substitutions in the wild-type protein, a strategy commonly used for α-helix disruption[53]. We substituted five residues in positions "a" of the heptad repeats with a single Pro (Fig. S8A). These substitutions are predicted by COILS to disrupt the CC domain significantly (Fig. S8B). Hereinafter we named this mutant as GFP-MED15PP.

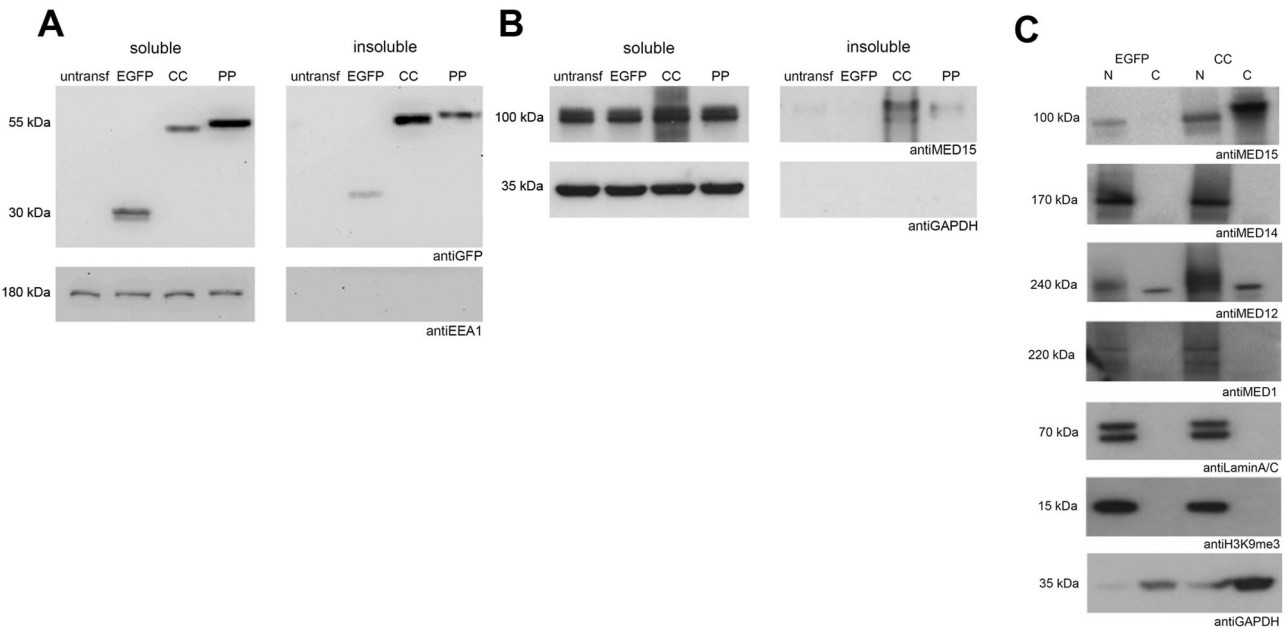

**Fig. 7 MED15-PrLD solubility and endogenous MED15 localization in HeLa cells.** Cell extracts of HeLa were processed for soluble examination of **A** EGFP, EGFP-MED15CC (CC), and EGFP-MED15PP (PP) or **B** endogenous MED15 by western blot in untransfected (untransf) cells or after expression of EGFP, EGFP-MED15CC (CC), or EGFP-MED15PP (PP) proteins. EEA1 and GAPDH were blotted as loading controls. **C** Cell extracts of HeLa were processed for nuclear/cytoplasmic (N/C) examination of endogenous MED15, MED14, MED12, and MED1 mediator subunits by western blot after expression of EGFP or EGFP-MED15CC (CC) proteins. LaminA/C, H3K9me3, and GAPDH were blotted as loading controls.

cellular inclusion-forming propensities, EGFP-MED15CC was localized mainly in the detergent-insoluble fraction upon HeLa cells fractionation, whereas EGFP-MED15PP remained mostly soluble (Fig. 7A).

Overall, the cellular data converge to indicate that, as it happens in vitro, the CC conformation mediates MED15-PrLD aggregation and the formation of protein inclusions in human cells.

We have shown that, in vitro, the MED15-PrLD seeds the amyloid formation of its soluble counterpart efficiently. This activity immediately suggested that, when expressed in cells, it may transmit its amyloid-like state and potentially seed the aggregation of the endogenous, soluble, and functional MED15 protein. Western blot and immunostaining of the endogenous full-length MED15 indicate that EGFP-MED15CC transient expression in HeLa cells results in a fraction of the otherwise soluble endogenous MED15 being recruited into the detergent-insoluble fraction (Fig. 7B). This pro-aggregation effect was reduced when we expressed the CC-disrupted EGFP-MED15PP variant instead (Fig. 7B).

Despite MED15 is assumed to be a nuclear protein, its presence in the cytosol has been described[24] and it has been suggested that this subunit might experiment indirect nuclear export/import through a piggyback mechanism[54]. We speculated that sequestration of the endogenous protein by aggregated MED15-PrLD should increase its levels in the cytosol. HeLa cells fractionation into cytoplasmic and nuclear fractions confirmed the expected redistribution of the endogenous protein upon EGFP-MED15CC expression (Fig. 7C). Mislocalization of endogenous MED15 in the cytosol might result from the retention of newly synthesized protein and/or by an alteration of the nucleo-cytoplasmic shuttling. In both cases, the interaction with aggregated cytosolic EGFP-MED15CC will difficult the transit to the nucleus.

Next, we assessed if the mislocalization of endogenous MED15 might also alter the distribution of other subunits of the Mediator complex, particularly MED14, MED12, and MED1, placed at the tail, CDK8, and middle modules, respectively. In contrast to MED15, none of these endogenous subunits mislocalizes upon

expression of EGFP-MED15CC (Fig. 7C), which indicates that as expected for a prion-like protein mechanism, cytoplasmic recruitment is sequence-specific.

Overall, the data in this section converge to indicate that MED15-PrLD possesses intracellular prion-like activity, related to the CC propensity of its Q-rich segment.

## Discussion

We have identified the presence of CC domains in a significant number of human PrLDs. This result extends the previous observation that these α-helical domains were recurrent in yeast prions[9], indicating that CC constitute a generic feature of many functional prion and prion-like proteins.

PrLDs are considered and consistently predicted to be disordered in their soluble states. Accordingly, a disordered to β-sheet transition is seen as the generic mechanism behind their aggregation, similar to pathogenic amyloids[3,4]. Instead, our data are consistent with CC domains playing a critical role in the structural dynamics of at least a fraction of human PrLDs. Accordingly, disrupting or destabilizing the MED15-PrLD CC, either genetically, by Pro mutations, or chemically, with urea, abolishes amyloid formation, whereas its stabilization with TFE accelerates aggregation.

To the best of our knowledge, this is the first time that the transition of a complete PrLD from an initial soluble CC conformation to an amyloid state has been characterized biophysically since previous reports involved relatively short peptides (17 residues) of the yeast Ure2p PrD[9]. These studies did not allow to discriminate if α-helical CC were self-sufficient mediators of the aggregation of functional prions or just intermediates or facilitators in the process of β-sheet formation. Our results support this second mechanism because the spontaneous aggregation of MED15-PrLD into amyloid fibrils involves a progressive conformational shift from an initial α-helical state to a β-sheet-rich structure, which occurs concomitantly with the aggregation of the domain.

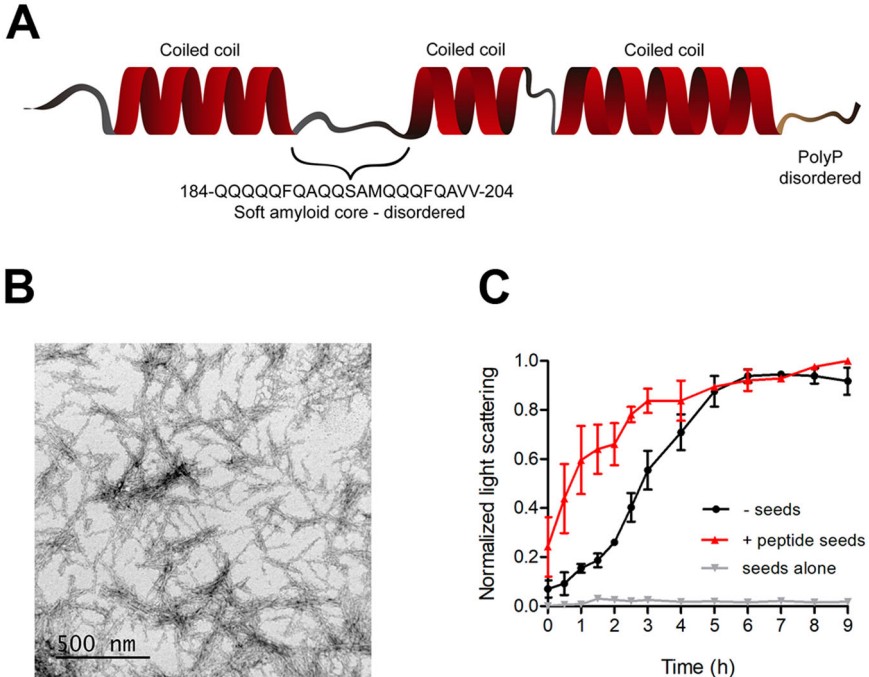

**Fig. 8 MED15 soft amyloid core seeds MED15-PrLD aggregation. A** Schematic diagram of MED15-PrLD structure. It consists of three coiled-coil regions predicted by COILS[33] (14-residue window) separated by disordered regions. The disorder region between the first and second coiled-coil corresponds to the soft amyloid core (SAC) predicted by pWaltz[73]. The final disordered region corresponds to the Pro residues flanking the polyQ tracts (polyP disordered). **B** Transmission electron microscopy image of aggregated 100 µM MED15CC SAC in 20 mM Tris pH 7.5 and 150 mM NaCl. **C** Light scattering kinetics of 5 µM GFP-MED15CC incubated at 37 °C in the absence or presence of 2% MED15 SAC seeds (peptide seeds). Kinetics of seeds alone is shown as control. Aggregation was performed in 20 mM Tris pH 7.5 and 150 mM NaCl.

The molecular mechanism by which a given CC converts into a β-sheet and forms amyloids is still unclear. In an elegant study, Kammerer et al.[55] designed a short peptide (17 residues), referred to as ccβ, that folds into a CC under ambient conditions but transforms into amyloid fibrils at elevated temperatures. This allowed the authors to trigger the CC to β-sheet transition in a controlled manner and to dissect the relative importance of the forces driving this conformational change. The study revealed that the nature of residues at position "f" of the heptad repeat, the most exposed region in a CC, was the most critical feature for the peptide transition. Hydrophobic residues at this position dramatically accelerated aggregation, whereas polar amino acids abolished it. These hydrophobic residues would facilitate multimerization, a reaction that depends on local contacts and would be favored at increased protein concentrations. Besides, a hydrophobic amino acid at position "f" would allow forming a continuous hydrophobic patch in a β-sheet, together with hydrophobic residues in positions "a" and "d", all lying in one side of the sequence in a fully extended conformation[56]. The same mechanism would likely apply for polyQ tracts, since Gln behaves as an ambivalent hydrophobe[41], with contacts becoming more favorable as the polyQ, and therefore the CC, expands.

Although our model system is far more complex, it shares significant features with the behavior of ccβ. In both cases, CD data indicated a transition from an initial α-helical to a β-sheet structure, the aggregated states corresponded to ordered amyloid fibrils, and conformational conversions were significantly accelerated by increased protein concentrations. Importantly, according to COILS, all residues at position "f" of the MED15-PrLD CC domain are either hydrophobic or Gln (Fig. 1B). The need for specific spatial positioning of exposed hydrophobic residues to allow multimerization might explain the need to maintain the integrity of CC for MED15-PrLD aggregation. An

essential difference between ccβ and MED15-PrLD is that the structural transition of the former requires native state destabilization by temperature, whereas in the second case, it occurs spontaneously at room temperature. This indicates that the energy barrier between the CC and amyloid states of MED15-PrLD is significantly lower than that of ccβ, something expected for a domain that should experiment conformational shifts under physiological conditions.

A CC-mediated model does not exclude the interplay of CCs with other PrLD elements in the aggregation process. The predicted CC regions in MED15-PrLD are discontinuous, with two short regions devoid of significant α-helical propensity linking them (Fig. 1B). We have shown that a majority of yeast PrDs contain sequence stretches able to assemble into highly ordered amyloid fibrils[57]. These regions differ from the classical amyloid cores of pathogenic proteins in that they are slightly longer and more polar, in such a way that the amyloid potential is less concentrated, allowing the containing PrD to remain soluble under most physiological conditions, while still keeping a certain amyloid propensity that might facilitate assembly in certain circumstances[57,58]. These soft amyloid cores (SAC) are necessary to sustain yeast prions conversion in human cells[59]. We recently identified and characterized similar regions within several human PrLDs, including that of MED15[60]. The SAC of MED15-PrLD comprises residues 184-QQQQQFQAQQSAMQQQFQAVV-204 and a peptide correspondent to this stretch spontaneously forms highly ordered, β-sheet-rich, amyloid fibrils[60]. This SAC corresponds almost precisely to the connecting region between the first and second CC in MED15-PrLD (Figs. 1B and 8A). This sequence is predicted to be devoid of any CC propensity by COILS and completely disordered by both FoldIndex and Espritz, and accordingly to be significantly exposed to the solvent. Thus, contacts between these accessible regions in different MED15-

PrLD molecules are possible. Indeed, when we formed fibrils of the 21-residues peptide (Fig. 8B) and added them (2%) to a solution of soluble GFP-MED15CC, they efficiently seeded its aggregation, indicating that this region was recognized in the soluble and dimeric protein (Fig. 8C). However, it is important to note that this amyloidogenic sequence alone is not sufficient to promote aggregation since this reaction does not occur in the absence of a previous CC conformation. Thus, either α-helical regions drive the initial MED15-PrLD multimerization, and intermolecular SAC contacts occur afterward, or both regions act in parallel to mediate aggregation. In both instances, the presence of the SAC would contribute to decrease the energy barrier for amyloid formation, respect that of a CC-only driven reaction. This leads us to propose a mechanism in which both the misfolding of short disordered regions to β-sheet conformations and CC multimerization cooperate to promote MED15-PrLD amyloid formation, unifying thus two apparently contradictory models for prion conversion. Such a cooperative model would explain why MED15-PrLD fibrillates spontaneously, even if its polyQ tracts are much shorter than those in triplet expansion genetic diseases[61].

We show that a dimeric CC structure is the thermodynamically dominant MED15-PrLD form in solution, immediately suggesting that MED15 might form homodimers spontaneously. No studies on the oligomeric state of full-length MED15 are available, but it has been demonstrated that yMED15 assembles into a dimer over a broad range of concentrations, suggesting that this state is also accessible to MED15[62]. The existence of functional MED15 as a dimeric protein stabilized by CC contacts is consistent with the observation that proteins containing CC are overrepresented among its interactors, since apart from homotypic interactions, CC also facilitate heterotypic PPI[63]. Human prion-like proteins containing CC at their PrLDs function in transcriptional regulation and comprise important coactivators, including NCOA2, MAML2, CBP, and MED12, all interconnected with MED15. As in humans, transcription regulators are prevailing in the yeast prion family[6], and they include four *bona fide* prions: Ure2p, Swi1, Cyc8, Mot3. Not surprisingly, Fiumara et al.[9] demonstrated that all these proteins contain CC at their PrDs. Thus, transcription appears to be an important cellular event that is influenced by prions and prion-like proteins across species, likely thanks to their ubiquitous CC domains. As it occurs in yeast, human prion-like transcription coactivators might permit the generation of phenotypic heterogeneity, allowing cells to adapt in front of intrinsic and extrinsic changes.

For most yeast prions, the new phenotype arising from their conversion to the amyloid state arises from a loss of function. In this way, Ure2p is a dimeric protein whose globular functional domain binds to the transcription factor Gln3p, preventing its migration to the nucleus[64]. This interaction is lost in the Ure2p prion amyloid state, allowing Gln3p to activate the transcription of a series of genes that were previously silent. However, the Ure2p globular domain is not misfolded in the process and maintains its original fold but becomes occluded from interaction and loses its original function[51]. Similarly, when MED15-PrLD accesses a highly ordered amyloid state, the contiguous GFP moiety remains folded and fluorescent, suggesting that a mechanism equivalent to that of Ure2p might regulate MED15 activity.

In vitro, preformed fibrils of MED15-PrLD can seed the aggregation of their soluble counterpart, implying that they facilitate a conformational conversion from an initial α-helical to a β-sheet structure. When MED15-PrLD is expressed in human cells, a significant fraction of the endogenous MED15 protein localizes to the detergent-insoluble cytoplasmic fraction suggesting that an analogous mechanism applies inside cells and thus

that MED15 can act in a prion-like manner in a physiological context. The ability to aggregate inside cells and to potentially transmit this conformation is, again, strongly dependent on the integrity of the N-terminal CC domain, as demonstrated by the reduced ability to form cellular inclusions and to interact with the endogenous protein of the MED15PP variant.

MED15 is part of the mediator tail module and is known to serve as a hub for signaling pathways. MED15 is overexpressed in different cancers and correlates with the clinical outcome and the recurrence of the disease[18–20,23,65]. The molecular mechanism by which increased levels of MED15 contributes to these malign phenotypes remains unknown, but it is assumed that they result from an increased and sustained transcriptional activation[19,20,23]. MED15-PrLD forms intracellular inclusions in different human cell lines, including laryngeal and oral squamous cell carcinoma (Fig. S10), two tumors with MED15 implication[19]. Under the assumption that, as it occurs for yMED15, the aggregation propensity of the PrLD is maintained in the context of the full-length protein, our results suggest alternative mechanisms to explain the association of this prion-like protein with cancer. Overexpression would increase the number of MED15 pre-associating N-terminal CC domains and thus their effective local concentration, facilitating both the establishment of CC homotypic and heterotypic interactions and PrLD conformational conversion.

MED15-PrLD mediated homotypic interactions would favor multimerization, and this might prevent the establishment of other relevant MED15 PPI, occluding the access to TFs interacting motifs, like those in the KIX domain, as it happens in yeast Ure2p[51]. Alternatively, the self-association of MED15 might result in enhanced formation of transcriptional hubs by phase separation, as described for the ENL chromatin reader[66]. Both situations are expected to result in widespread transcriptional changes. Because a significant fraction of MED15 interactors contains CC domains, increased heterotypic interactions and subsequent multimerization by side-to-side CC interactions, or from the swapping between protomers, are expected to exert a similar effect to homotypic contacts, either decreasing or exacerbating the interactors function. Another possibility is that the MED15-PrLD would establish promiscuous interactions with other Q-rich CC-containing proteins that are not usual MED15 partners, sequestering them. Since many of these proteins are expected to be TFs, these degenerated CC interactions would significantly impact transcription and signaling pathways, as demonstrated for polyQ-expanded proteins in neurodegenerative diseases[11,67]. In extreme cases, the MED15-PrLD might template and enhance the transition of other Q-rich proteins towards the formation of toxic amyloid aggregates, as previously shown for human ATXN1[40]. Deciphering if any of these mechanisms or a combination of them is behind MED15 association with different cancers might offer novel therapeutic avenues for their treatment.

## Methods

**Bioinformatics analysis.** Human reviewed proteins from the standard proteome (Proteome ID UP000005640, release 2019_06) were downloaded from UniProtKB/Swiss-Prot[68,69]. MED15 mouse interactors were obtained from[42]. Their human homologous were obtained by blasting each sequence against their human UniProtKB/Swiss-Prot counterparts. For each protein, the highest sequential candidate (>75% homology) was obtained.

Human reviewed proteins were screened with PLAAC algorithm[32] using as background probability the precompiled frequencies of human proteome to obtain the list of putative proteins with PrLDs.

Coiled-coil prediction was performed using the C version of COILS[33] or PairCoil2[70] with window length of 21 residues, unless is otherwise indicated.

Prion-like proteins and MED15 interactors datasets were analyzed for enrichment with the Functional Annotation Tool of DAVID 6.8 (Database for Annotation, Visualization and Integrated Discovery)[34] and ordered by p-value. The default "Direct GO" category was selected for the three ontologies: molecular function, biological process and cellular component. Direct GO uses mappings

directly provided by the source annotation, thus minimizing redundant parent terms or very high-level classes.

**MED15 sequences**. MED15-PrLD sequence (residues 141–316) was obtained from Erich E. Wanker research group and fused to GFP sequence in pET21b plasmid. GFP-MED15CC sequence is shown below with MED15-PrLD sequence underlined.

```
MASMTGGQQMGRDPNSSKGEELFTGVVPILVELDGDVNGHKF
SVSGEGEGDATYGKLTLKFICTTGKLPVPWPTLVTTLTYGVQCF
SRYPDHMKRHDFFKSAMPEGYVQERTISFKDDGNYKTRAEVKFE
GDTLVNRIELKGIDFKEDGNILGHKLEYNYNSHNVYITADKQKN
GIKANFKIRHNIEDGSVQLADHYQQNTPIGDGPVLLPDNHYLST
QSALSKDPNEKRDHMVLLEFVTAAGITHGMDELYKKLLEVLFQG
PMAVVSTATPQTQLQLQQVALQQQQQQQQFQQQQQQAALQQQQQQ
QQQQQFQAQQSAMQQQFQAVVQQQQQLQQQQQQQQQLQQQQ
QQQQIQQQQQQLQRIAQLQLQQQQQQQQQQQQQQQQQALQAQPPIQ
QPPMQQPQPPPSQALPQQLQQMHHTQHHQPPPQPQQPPVAQNQP
SAAALEHHHHHH
```

MED15 PP sequence was purchased from Genscript as GFP-MED15PP fusion in a pET21b plasmid. The sequence is the same as GFP-MED15CC except 5 Pro substitutions in MED15 CC region (Fig. S6A).

**Protein expression and purification**. In all, 100 ml of Luria Broth (LB) with 100 µg/ml ampicillin (amp) and 34 µg/ml chloramphenicol (clm) were inoculated by single colony of BL21 Rosetta cells with pET21b-GFP-MED15CC/PP and incubated overnight at 37 °C and 250 rpm. In total, 25 ml of saturated overnight culture was transferred into 1L LB-amp-clm and incubated at 37 °C and 250 rpm. Protein expression was induced at $OD_{600} = 0.5$ by addition of isopropyl-β-D-thiogalacto-pyranoside (IPTG) to a final concentration of 0.5 mM. The induced culture was incubated O/N at 20 °C and 250 rpm. Cells were then harvested by centrifugation for 15 min at 4000×$g$ and 4 °C (Beckman Coulter™ Avanti Centrifuge J-26XPI). The cell pellet was frosted at −80 °C.

Pellets from 2L cell culture were resuspended in 30 ml binding buffer (20 mM TrisHCl pH 8, 500 mM NaCl, 20 mM imidazole, 10% glycerol) supplemented with 0.2 mg/ml lysozyme, 20 µg/ml DNase, and 1 tablet of protease inhibitor cocktail EDTA free (Roche). The suspension was incubated for 20 min at 4 °C with slow agitation and then lysed by 5 min sonication on ice (Branson Digital Sonifier). Lysate cells were centrifuged for 30 min at 30,000×$g$ and 4 °C. The supernatant was filtered through a 0.45-µm PVDF membrane and loaded onto a HisTrap FF Ni-column (GE Healthcare) at a flow rate of 4 ml/min. Protein was eluted by one-step procedure using elution buffer (20 mM TrisHCl pH 8, 500 mM NaCl, 250 mM imidazole, and 10% glycerol). His purified protein was treated with 0.2 mg/ml RNase (Thermo Scientific) for 15 min at 37 °C and 1 mM PMSF was added to avoid protein degradation. Native buffer (20 mM TrisHCl pH 8, 500 mM NaCl, and 10% glycerol) was added to the protein sample to reduce imidazole to 20 mM and a second His trap step was performed as before. Then, protein was filtered using 0.22 µm PVDF membrane, concentrated using 10 K Amicon (Millipore) to 2.5 ml, filtered again and subjected to PD10 column equilibrated in 20 mM TrisHCl pH 7.5, 500 mM NaCl, 10% glycerol. Purified protein was concentrated to 400 µM, filtered and stored in small volume aliquots at −80 °C. Protein concentration and absence of RNA was confirmed by 260 nm and 280 nm absorbance values using Specord® 200 Plus spectrophotometer (Analyticjena).

GFP was purified as described in Gil-Garcia et al.[71] and Sup35NM was purified as described in Sant'anna et al.[57].

**Size exclusion chromatography**. In all, 2 mg/ml of purified protein was filtered with 0.22 µm PVDF membrane and loaded on a Superdex 200 HR 10/30 (GE Healthcare) equilibrated with 20 mM TrisHCl pH 7.4, 150 mM NaCl, and 10% glycerol. The fractions were analyzed by 12% SDS-PAGE.

**Native page**. In all, 0.5 mg/ml of purified protein was loaded in a 12.5% blue native polyacrylamide gel electrophoresis for the analysis of protein oligomers.

**Dynamic light scattering**. GFP and GFP-MED15CC protein size was determined using a Malvern Zetasizer Nano Series (Malvern instruments, UK) in 20 mM TrisHCl pH 7.5 and 150 mM NaCl at different protein concentrations per duplicate. Protein diameter was extracted from volume size.

**GFP absorption**. GFP absorbance was monitored from 400 to 600 nm using a Specord® 200 Plus spectrophotometer (Analyticjena).

**Protein aggregation**. Right before each experiment, the stock solutions were diluted to 5 µM in 20 mM TrisHCl pH 7.4 and 150 mM NaCl. For aggregation assays the samples were incubated at 37 °C without agitation. In some experiments, protein aggregation was performed in the presence of 2 M urea or 5% 2,2,2-trifluoroethanol (TFE) (Fluka). All aggregation experiments were performed per triplicate.

**GFP fluorescence**. In all, 50 µl sample was centrifuged for 5 min at max speed and RT. Then, supernatant fraction was used for GFP fluorescence measurements unless indicated in the text. GFP fluorescence was monitored using a JASCO Spectrofluorometer FP-8200. The conditions of the spectra acquisition were: excitation wavelength of 485 nm, emission range from 500 to 600 nm, slit widths of 5 nm, 0.5 nm interval, 1 s response, and 1000 nm/min scan rate. The supernatant fraction was also analyzed by 12% SDS-PAGE.

**Synchronous light scattering**. Synchronous light scattering was monitored using a JASCO Spectrofluorometer FP-8200. The conditions of the spectra acquisition were: excitation wavelength of 330 nm, emission range from 320 to 340 nm, slit widths of 5 nm, 0.5 nm interval, 1 s response, and 1000 nm/min scan rate.

**Congo Red precipitation**. CR 200 µM in $H_2O$ was centrifuged before the experiment to remove any possible precipitate. Buffer, GFP, GFP-MED15CC, or Sup35NM aggregates were mixed 1:1 (v/v) with CR 200 µM. Samples were incubated for 1 h at RT covered from light. Then, we centrifuged the samples for 5 min at max speed and RT. Appearance of a red pellet was indicative of the presence of amyloid fibrils.

**Circular Dichroism**. CD experiments were performed using a JASCO J-715 spectropolarimeter. Measurements of the far-UV CD spectra (260–200 nm) were made by the addition of 200 µl of 5 µM sample to a cuvette of 0.1 cm path-length. Spectra were recorded at room temperature, 1 nm band width, and 100 nm/min scan rate. The resulting spectrum was the average of 5 scans. The contribution of the buffer was subtracted.

**Fourier transform infrared spectroscopy**. In all, 100 µl aggregated protein was centrifuged and washed one time with $H_2O$ to remove the presence of salts. The final pellet was resupended in 10 µl $H_2O$. FTIR experiments were performed using a Bruker Tensor 27 FT-IR spectrometer (Bruker Optics Inc) with a Golden Gate MKII ATR accessory. Each spectrum consists of 32 independent scans, measured at a spectral resolution of 4 cm$^{-1}$ within 1800–1500 cm$^{-1}$ range. All spectral data were acquired and normalized using the OPUS MIR Tensor 27 software. Data was afterwards deconvoluted using the Peak Fit 4.12 program.

**Transmission electron microscopy**. The morphology of the aggregated proteins was evaluated by negative staining and using a JEOL TEM-1400Plus Transmission Electron Microscope, 80 KV. 5 µM aggregated protein solution was diluted to 1 µM final concentration in $H_2O$. In total, 5 µl of the diluted solution was placed on carbon-coated copper grids and incubated for 5 min. The grids were then washed and stained with 5 µl of 2% w/v uranyl acetate for 1 min. Then, grids were washed again before analysis.

**Mammalian molecular cloning**. MED15 CC and PP sequence were amplified by PCR from pET21b-GFP-MED15CC/PP and assembled to pEGFP-C3 (Clontech, plasmid #6082–1) HindIII and BamHI digested plasmid using the NEBuilder HiFi DNA Assembly Master Mix (New England Biolabs). EGFP and EGFP-MED15CC were amplified by PCR from pEGFP-C3 and pEGFP-C3-MED15CC, respectively, and assembled to pTetO-FUW-OSKM (Addgene, plasmid #20321) EcoRI digested plasmid using the NEBuilder HiFi DNA Assembly Master Mix (New England Biolabs). The final vectors were transformed into XL1Blue chemically competent E. coli cells.

**Mammalian cell culture**. HeLa cells were grown and maintained in MEMα Glutamax medium (Thermo Fisher Scientific) supplemented with 10% fetal bovine serum (FBS) (Thermo Fisher Scientific) at 37 °C and 5% $CO_2$. HEK-293T were grown and maintained in DMEM medium supplemented with 10% FBS (Gibco) with antibiotics (penicillin/streptomycin 100 U/ml) on gelatin-coated plates at 37 °C and 5% $CO_2$. Laryngeal HNSCC (UT-SCC-42B) and oral HNSCC (UT-SCC-2) were grown and maintained in DMEM medium (Thermo Fisher Scientific) supplemented with 10% fetal bovine serum (FBS) (Thermo Fisher Scientific) at 37 °C and 5% $CO_2$. Cells were passaged and plated using 1x TrypLE Express (Thermo Fisher Scientific) or Trypsin-EDTA (0.05% Trypsin 0.53 mM EDTA•4Na) liquid (Invitrogen).

**Production of lentivirus and infection of recipient cells**. Lentiviral supernatants were produced in HEK-293T cells ($5 \times 10^6$ cells per 100 mm diameter dish). Vector transfections were performed using Fugene-6 transfection reagent (Roche) according to the manufacturer's protocol.

For lentiviral production, per dish, 293T cells were transfected with 3 plasmids: (i) the ecotropic lentiviral envelope packaging plasmid pMD2.G (0.3 µg; Addgene, plasmid #12259; containing the VsVg gene); (ii) the lentiviral packaging plasmid pCMV-dR8.91 (3.0 µg); (from: Harvard Medical School, plasmid #516); (iii) plus one of the following lentiviral expression constructs (3.0 µg) expressing either the FUW-M2-rtTA vector (Addgene #20342) or, pTetO-FUW-EGFP-MED15CC or pTetO-FUW-EGFP.

Two days later, viral supernatants (10 ml) were collected serially during the subsequent 48 h, at 12-h intervals, each time adding fresh medium to the cells (10

ml). The recipient 293T cells were seeded the previous day ($1.5 \times 10^5$ cells per well in a six-well plate) and each well received 1.0 ml of the corresponding lentiviral supernatants. This procedure was repeated every 12 h for 2 days (a total of 3 additions). Recipient cells received both the FUW-M2-rtTA plus either pTetO-FUW-EGFP-MED15CC or pTetO-FUW-EGFP.

In all, 24 h after lentiviral infection was completed; lentiviral expressing cells were used directly. Cell samples were tested for expression of EGFP or EGFP-MED15CC by immunofluorescence 72 h after addition of 1 µg/ml doxycycline.

**Mammalian cells transfection.** HeLa, laryngeal HNSCC (UT-SCC-42B) and oral HNSCC (UT-SCC-2) cells were transfected 24 h after seeding, on eight-well glass slides for immunofluorescence or on 10 cm dish for western blot, with 400 ng DNA using Lipofectamine2000 (Invitrogen). After 4 h of transfection, we changed cellular media to fresh one.

**Immunofluorescence.** Cells were fixed with 4% parafolmaldehyde (141451, Pan-ReacAppliChem), permeabilized with 0.2% TritonX100 and blocked with 1% bovine serum albumin (BSA) or 5% filtered FCS (Foetal Calf Serum). Primary antibody used was against MED15 (11566-1-AP, Proteintech). For visualization, the appropriate host-specific Alexa Fluor 555 (Thermo Fisher Scientific) secondary antibody was used. Slides were mounted using ProLong Gold antifade reagent with DAPI (Invitrogen). Images were captured using a Leica TCS SP5 confocal microscope (Leica Biosystems) with a ×63 oil objective.

**Soluble/insoluble fractionation.** Transfected HeLa cells were trypsinized, washed with PBS1x and resuspended in 0.5 ml RIPA buffer (50 mM Tris pH 7.4, 150 mM NaCl, 1% TritonX100, 2 mM EDTA, 0.5% sodium deoxycholate, and 0.1% SDS) supplemented with protease inhibitors. After 10 min incubation on ice, cells were centrifuged for 10 min at max speed and 4 °C. The supernatant corresponding to the soluble fraction was placed in a new eppendorf tube. The pellet corresponding to the insoluble fraction was washed with RIPA supplemented with protease inhibitors, resuspended in 0.5 ml urea buffer (30 mM Tris pH 8.5, 8 M urea, 2% CHAPS) and incubated for 20 min at RT and agitation. Protein concentration of the soluble fraction was measured by Bradford and 20 µg were loaded for western blot analysis. We loaded the same volume for the insoluble fraction. Primary antibody used was against MED15 (11566-1-AP, Proteintech), GFP (G6795, Sigma), GAPDH (AM4300, Invitrogen), and EEA1 (610457, BD Biosciences). The appropriate host-specific HRP conjugated secondary antibody (BioRad) was used. Western Blot was revealed using Immobilon® Forte Western HRP substrate (Millipore) in a VersaDoc (BioRad).

**Nuclear/cytoplasmic fractionation.** Nuclear and cytoplasmic fractionation was performed with the NE-PER kit (Thermofisher #78833). In all, 10 µg protein was loaded for western blot analyses. Primary antibody used was against MED15 (11566-1-AP, Proteintech), GFP (ab13970, Abcam), MED1 (51613, Cell signaling), MED14 (ab72141, Abcam), MED12 (A300-774A, Bethyl Laboratoris), H3K9me3 (2118 s, Cell Signaling), and LaminA/C (sc-6215 (N-18) or sc-376248 (E-1), Santa Cruz Biotechnology).

**Statistics and reproducibility.** Statistical analyses were performed using GraphPad Prism v8 or SigmaPlot v10. Unless otherwise indicated data are shown as the mean ± standard deviation (SD). In Fig. 6C the data correspond to the mean ± SEM and significance in difference between two groups were tested by Student *t* test. Triplicate samples were analyzed per experiment.

**Reporting summary.** Further information on research design is available in the Nature Research Reporting Summary linked to this article.

## Data availability
The unprocessed immunoblotting images are shown in Supplementary Fig. 11. The source data behind the graphs are provided in Supplementary Data 3. All datasets in the current study are available from the corresponding author on reasonable requests.

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

## Acknowledgements

We are grateful to UAB microscopy service of Barcelona for their technical advice. We thank Erich E. Wanker research group for providing us GST-MED15CC plasmid. We thank Marcos Gil for providing us GFP purified protein. I.C. was funded by Secretaria d'Universitats i Recerca de la Generalitat de Catalunya and European Social Fund. Work in the laboratory of M.S. was funded by the IRB, and by the Spanish Ministry of Science co-funded by European Regional Development Fund (ERDF; SAF2017-82613-R), the European Research Council (ERC-2014-AdG/669622), "la Caixa" Foundation, and Secretaria d'Universitats i Recerca del Departament d'Empresa i Coneixement de Catalonia (Grup de Recerca consolidat 2017 SGR 282). S.V. acknowledges funding from MINECO (BIO2016–78310-R) and ICREA (ICREA-Academia 2016). C.B. acknowledges funding from "Ministerio de Educación y Formación Profesional".

## Author contributions

C.B. designed and performed the experiments, analyzed the data, and wrote the manuscript. I.C, C.L., and M.G-G. contributed to specific experiments, V.I. performed the bioinformatics analysis. M.S. designed the stable cell line experiments. V.I., I.C., C.L., and M.S. edited the manuscript. S.V. designed the project, analyzed the data, and wrote the manuscript.

## Competing interests

The authors declare no competing interests.
