## [Peer Review File · Communications Biology]

Reviewers' comments:

Reviewer #1 (Remarks to the Author):

In this manuscript, Batlle et al. identified by computational means the presence of coiled-coil (CC) domains that overlap with polyQ tracts in human proteins containing prion-like domains (PrLDs). They demonstrate convincingly that human MED15 Mediator complex subunit forms homodimers in solution mediated by coiled-coil interactions and that MED15CC aggregates into amyloid fibrils. Overall, the purpose of the study is well defined, the manuscript is well written, experiments are appropriate and the conclusions support the results shown. However some data and aspects of the manuscript are worth clarifying.

Major comments

1. The experimental data (concentration and incubation times) on the aggregation kinetics are not uniform throughout the manuscript. For example it is not clear to me why protein aggregation was performed during 4 days in the Congo red experiment (Figure 3E) when transmission electron microscopy was done at t=9h (Figure 3F) in accordance with the plateau phase (Figure 4). Please be also consistent throughout the manuscript in term of protein concentration. Sometimes μM is used, sometimes mg/ml. In particular for Figure 3E again, it is said that 1mg/ml of GFP was used. If I am not mistaken, it corresponds to a concentration of $37\mu\text{M}$, well above the $5\mu\text{M}$ used in the other experiments.
2. There are also discrepancies in the cellular experiments (stably HEK293T t=72h versus HeLa cells transient t=24h). Do the authors already see intracellular inclusions after 24h in HEK293T? If possible, it would be interesting to show more cells for HeLa cells in the field of view. With only one cell with EGFP-MED15CC (Figure 6B), it is difficult to appreciate the level of aggregation.
3. In Figure 7, when EGFP-MED15CC mislocalizes endogenous MED15, will other Mediator complex subunits be altered? This should be examined. In the nuclear/cytoplasmic fractionation (Figure 7C), the cytoplasmic MED15 band appears to migrate slightly faster than the nuclear band. Please comment. Was it known that MED15 may actually shuttle between the nucleus and the cytoplasm?
4. In the current version, the supplementary tables are not friendly to read and it is hard to know which one is which. Maybe a landscape version (versus portrait) could be better. Because many identified proteins are transcriptional coactivators (CBP, TBP, FOX2, etc...), a supplementary figure (like Figure 1A) showing the location of predicted coiled-coils in these proteins will help the reader.

Minor comments

1. In the abstract and introduction (page 3, fourth paragraph), 'globular domains adjacent to MED15-PrLD...' should be 'GFP globular domain' given that only the GFP globular integrity is evaluated in the manuscript.
2. Materials & Methods page 22. 'Purified protein was concentrated to 0.5ml'. Please provide concentration in mg/ml and/or in μM .
3. In the references section, page and/or volume are missing for some references (for example reference 26, 53, 58).

4. Figure legends. Figure 1 page 18. Please described briefly in the legends all the algorithms that were used. The schematic diagram of the heptad repeats is not clear to follow. Please provide amino acid limits for each heptad.

5. Figure 3. The superposition of curves are sometimes not easy to see (Figure 3C). It will be maybe easier to follow with a split figure, 1 part with GFP and one part with GFP-MED15CC.

Reviewer #2 (Remarks to the Author):

In this manuscript, "MED15 prion-like domain forms a coiled-coil responsible for its amyloid conversion and propagation" by Batlle et al., the authors describe a fairly extensive biophysical analysis of a prion-like domain in MED15. They build on previous work to demonstrate that a coiled-coil domain contributes to homodimerization and aggregation of the Gln-rich region of the protein.

Overall, the authors provide a compelling set of experiments that support their claims. Also, the data support a novel model that impacts our understanding of protein aggregation diseases.

My only major concern with the manuscript is that the authors rely on a GFP-fusion for all experiments. While the proline mutations are a reasonable negative control, they do not eliminate the possibility that GFP contributes to the aggregation propensity. I recommend that they attempt the aggregation studies with at least one alternative fusion partner (excluding GST, which dimerizes itself).

Reviewer #1 (Remarks to the Author):

In this manuscript, Battle et al. identified by computational means the presence of coiled-coil (CC) domains that overlap with polyQ tracts in human proteins containing prion-like domains (PrLDs). They demonstrate convincingly that human MED15 Mediator complex subunit forms homodimers in solution mediated by coiled-coil interactions and that MED15CC aggregates into amyloid fibrils. Overall, the purpose of the study is well defined, the manuscript is well written, experiments are appropriate and the conclusions support the results shown. However some data and aspects of the manuscript are worth clarifying.

Authors: We thank the reviewer for his/her positive opinion on our work.

Major comments

1. The experimental data (concentration and incubation times) on the aggregation kinetics are not uniform throughout the manuscript. For example it is not clear to me why protein aggregation was performed during 4 days in the Congo red experiment (Figure 3E) when transmission electron microscopy was done at $t=9$ h (Figure 3F) in accordance with the plateau phase (Figure 4). Please be also consistent throughout the manuscript in term of protein concentration. Sometimes μ M is used, sometimes mg/ml. In particular for Figure 3E again, it is said that 1mg/ml of GFP was used. If I am not mistaken, it corresponds to a concentration of 37μ M, well above the 5μ M used in the other experiments.

Authors: We agree with the reviewer respect the Congo red (CR) experiment. It was performed after 4 days and at 1 mg/ml because these were the conditions we use in the lab to measure Sup35NM aggregation, which served as a positive control.

We have repeated the CR experiment at $t= 9$ h and 5μ M for all the samples (except the Sup35NM control which requires 10μ M) confirming the amyloid-like nature of incubated GFP-MED15CC, which like Sup35NM, binds to CR, in contrast to GFP alone or the GFP-MED15PP mutant that is now incorporated to the analysis (new Figure 3E). All concentrations are now indicated as μ M in the results section; for some specific cases in the methods section, we kept mg/ml since we thought they were more informative.

2. There are also discrepancies in the cellular experiments (stably HEK293T $t=72$ h versus HeLa cells transient $t=24$ h). Do the authors already see intracellular inclusions after 24h in HEK293T? If possible, it would be interesting to show more cells for HeLa cells in the field of view. With only one cell with EGFP-MED15CC (Figure 6B), it is difficult to appreciate the level of aggregation.

Authors: They are good observations. In HEK293T after 24 hours of induction we already see cytoplasmic MED15 inclusions. At this early timepoint, we observe that the inclusions are distributed less heterogeneously in the cytoplasm compared to later timepoints, this is now included in page 10 and Figure 6. We also include as supplementary Figure S9B images containing multiple HeLa cells in the field of view, for EGFP-MED15CC. In addition, for HeLa cells we also include an experiment using the TetO inducible system which provides information consistent with the constitutive

system were the presence of aggregates is already observable at 24 h and massive at 72 h. This is now included in Figure S9A and page 10.

3. In Figure 7, when EGFP-MED15CC mislocalizes endogenous MED15, will other Mediator complex subunits be altered? This should be examined. In the nuclear/cytoplasmic fractionation (Figure 7C), the cytoplasmic MED15 band appears to migrate slightly faster than the nuclear band. Please comment. Was it known that MED15 may actually shuttle between the nucleus and the cytoplasm?

Authors: This is a very interesting experiment to perform. We have assessed if the following subunits of the Mediator complex mislocalize: MED14, MED12 and MED1, placed at the tail, CDK8 and middle modules, respectively. No one show altered distribution when EGFP-MED15CC is expressed (New Figure 7C), indicating that we are catching a MED15 specific effect.

The fact that the two bands have slightly different mobility might respond to posttranslational modifications, but since we do not have evidences for that, we preferred to skip this discussion in the manuscript.

The presence of cytoplasmic MED15 has been previously described in the literature:

Klümper N., Syring I., Offermann A., Shaikhibrahim Z., Vogel W., Müller S.C., Ellinger J., Strauß A., Radzun H.J., Ströbel P., et al. Differential expression of Mediator complex subunit MED15 in testicular germ cell tumors. *Diagn. Pathol.* 2015;10:165. doi: 10.1186/s13000-015-0398-6

Indeed, both cytoplasm and nucleus appear as sub-cellular localization and place for expression for MED15 in Uniprot and the Human protein atlas (although the nuclear localization is predominant):

https://www.uniprot.org/uniprot/Q96RN5#subcellular_location

<https://www.proteinatlas.org/ENSG00000099917-MED15>

Finally, the phenomenon of “indirect” or “piggybacking” nuclear import or export is well known. Many proteins lacking NES are actually exported because they are bound to a partner that contains a NES. This mechanism has been suggested to apply for MED15 and other subunits of Mediator:

Tessier TM, MacNeil KM, Mymryk JS. Piggybacking on Classical Import and Other Non-Classical Mechanisms of Nuclear Import Appear Highly Prevalent within the Human Proteome. *Biology (Basel)*. 2020;9(8):188. doi:10.3390/biology9080188

This information has been included in the revised manuscript (Page 11).

4. In the current version, the supplementary tables are not friendly to read and it is hard to know which one is which. Maybe a landscape version (versus portrait) could be better. Because many identified proteins are transcriptional coactivators (CBP, TBP, FOXP2, etc...), a supplementary figure (like Figure 1A) showing the location of predicted coiled-coils in these proteins will help the reader.

Authors: We have made a new Figure S1, where the prion-like domains and coiled-coil regions are indicated for the different identified proteins.

The fact that the supplementary tables are not friendly comes from the fact that the system unavoidably converts the original excel files to PDFs and attaches them to the merged manuscript. In principle, both the reviewers and the readers would have access to the original excel files and these ugly automatically converted files would not be present in the published article.

Minor comments

1. In the abstract and introduction (page 3, fourth paragraph), 'globular domains adjacent to MED15-PrLD...' should be 'GFP globular domain' given that only the GFP globular integrity is evaluated in the manuscript.

Authors: we agree, it has been corrected.

2. Materials & Methods page 22. 'Purified protein was concentrated to 0.5ml'. Please provide concentration in mg/ml and/or in.

Authors: Thanks, it has been corrected to 400 μ M.

3. In the references section, page and/or volume are missing for some references (for example reference 26, 53, 58).

Authors: Thanks, they have been corrected.

4. Figure legends. Figure 1 page 18. Please described briefly in the legends all the algorithms that were used. The schematic diagram of the heptad repeats is not clear to follow. Please provide amino acid limits for each heptad.

Authors: Thanks, the algorithms have been added to the legend and heptad limits have been incorporated in Figure 1.

5. Figure 3. The superposition of curves are sometimes not easy to see (Figure 3C). It will be maybe easier to follow with a split figure, 1 part with GFP and one part with GFP-MED15CC.

Authors: We agree that in the specific case of the CD data (Figure 3C) it was better to split the graph and we have included in the main figure only the GFP-MED15CC data, generating a supplementary Figure S5 with the CD data of GFP alone.

Reviewer #2 (Remarks to the Author):

In this manuscript, "MED15 prion-like domain forms a coiled-coil responsible for its amyloid conversion and propagation" by Batlle et al., the authors describe a fairly extensive biophysical analysis of a prion-like domain in MED15. They build on

previous work to demonstrate that a coiled-coil domain contributes to homodimerization and aggregation of the Gln-rich region of the protein.

Overall, the authors provide a compelling set of experiments that support their claims. Also, the data support a novel model that impacts our understanding of protein aggregation diseases.

Authors: We thank the reviewer for his/her positive opinion on our work.

My only major concern with the manuscript is that the authors rely on a GFP-fusion for all experiments. While the proline mutations are a reasonable negative control, they do not eliminate the possibility that GFP contributes to the aggregation propensity. I recommend that they attempt the aggregation studies with at least one alternative fusion partner (excluding GST, which dimerizes itself).

Authors: As the reviewer states, both the GFP alone and the MED15CC proline mutant data argue that the MED15 prion-like domain is responsible for the observed phenotypes. However, we agree with the reviewer that the experiment he/she proposes makes sense.

Following the reviewer's suggestion, we have fused MED15CC independently to two different partners: SUMO and the Z-domain. We usually express fusions of different proteins to these small partners at high yields (Batlle et al. *Cell Rep.* 2020 **30**:1117-1128 / Wang et al. *Biomacromolecules.* 2020 **21**:4302-4312), and we know that they do not interfere with the fused protein's aggregation behavior.

However, despite the best of our efforts, SUMO-MED15CC and Z-domain-MED15CC fusions failed to express detectable levels and therefore could not be purified. We have tried a wide range of induction/repressing conditions and bacterial expression strains, including a strain over-expressing chaperones, without any success. We do not find any obvious explanation for this, except the intrinsic difficulty to express sequences with long poly-Q sequences. The expression of MED15CC alone was also unsuccessful.

Therefore, although we agree those experiments would have been a good control, we could not carry them out.

REVIEWERS' COMMENTS:

Reviewer #1 (Remarks to the Author):

The authors have addressed the critical comments in a satisfactory manner.

Reviewer #2 (Remarks to the Author):

For this revised manuscript, the authors attempted to address my primary concern that the GFP fusion influenced the study results. Unfortunately, they could not produce sufficient quantities of alternative fusion proteins (SUMO or Z-domain) or the isolated MED15CC. While it is intriguing that these alternative constructs did not express well, the heterologous expression of proteins can be very challenging and idiosyncratic.

Given that the authors attempted to address this issue in good faith and the quality of their studies otherwise, I feel that my concerns have been adequately addressed.